# Association between ethnic background and COVID-19 morbidity, mortality and vaccination in England: a multistate cohort analysis using the UK Biobank

Tomás Urdiales [1,2,3] Francesco Dernie [3] Martí Català,[3] Albert Prats-Uribe [3] Clara Prats,[1] Daniel Prieto-Alhambra [3]

CP and DP-A are joint senior authors.

[1]Department of Physics, Universitat Politècnica de Catalunya, Barcelona, Spain
[2]Department of Energy Technology, Royal Institute of Technology, Stockholm, Sweden
[3]Pharmaco- and Device Epidemiology, Centre for Statistics in Medicine, Nuffield Department of Orthopaedics, Rheumatology and Musculoskeletal Sciences, University of Oxford, Oxford, UK

**Correspondence to**
Dr Albert Prats-Uribe;
albert.prats-uribe@ndorms.ox.ac.uk and
Mr Tomás Urdiales;
tomas.urdiales@gmail.com

## ABSTRACT

**Objectives** Despite growing evidence suggesting increased COVID-19 mortality among people from ethnic minorities, little is known about milder forms of SARS-CoV-2 infection. We sought to explore the association between ethnic background and the probability of testing, testing positive, hospitalisation, COVID-19 mortality and vaccination uptake.

**Design** A multistate cohort analysis. Participants were followed between 8 April 2020 and 30 September 2021.

**Setting** The UK Biobank, which stores medical data on around half a million people who were recruited between 2006 and 2010.

**Participants** 405 541 subjects were eligible for analysis, limited to UK Biobank participants living in England. 23 891 (6%) of participants were non-white.

**Primary and secondary outcome measures** The associations between ethnic background and testing, testing positive, hospitalisation and COVID-19 mortality were studied using multistate survival analyses. The association with single and double-dose vaccination was also modelled. Multistate models adjusted for age, sex and socioeconomic deprivation were fitted to estimate adjusted HRs (aHR) for each of the multistate transitions.

**Results** 18 172 (4.5%) individuals tested positive, 3285 (0.8%) tested negative and then positive, 1490 (6.9% of those tested positive) were hospitalised, and 129 (0.6%) tested positive at the moment of hospital admission (ie, direct hospitalisation). Finally, 662 (17.4%) died after admission. Compared with white participants, Asian participants had an increased risk of negative to positive transition (aHR 1.24 (95% CI 1.02 to 1.52)), testing positive (95% CI 1.44 (1.33 to 1.55)) and direct hospitalisation (1.61 (95% CI 1.28 to 2.03)). Black participants had an increased risk of hospitalisation following a positive test (1.71 (95% CI 1.29 to 2.27)) and direct hospitalisation (1.90 (95% CI 1.51 to 2.39)). Although not the case for Asians (aHR 1.00 (95% CI 0.98 to 1.02)), black participants had a reduced vaccination probability (0.63 (95% CI 0.62 to 0.65)). In contrast, Chinese participants had a reduced risk of testing negative (aHR 0.64 (95% CI 0.57 to 0.73)), of testing positive (0.40 (95% CI 0.28 to 0.57)) and of vaccination (0.78 (95% CI 0.74 to 0.83)).

**Conclusions** We identified inequities in testing, vaccination and COVID-19 outcomes according to ethnicity in England. Compared with whites, Asian participants

## STRENGTHS AND LIMITATIONS OF THIS STUDY

⇒ This study uses a large cohort of adults, allowing for high-powered assessment of associations between baseline characteristics and COVID-19 outcomes.
⇒ Key confounders, both individual level and ecological, are adjusted for in our statistical analysis.
⇒ Our multistate analyses are able to take into account differential confounding structure for each transition.
⇒ The UK Biobank cohort is not optimally representative, limiting extrapolation of findings to the wider population.

had increased risks of infection and admission, and black participants had almost double hospitalisation risk, and a 40% lower vaccine uptake.

## INTRODUCTION

Since the first cases of human infection with SARS-CoV-2 were reported in Wuhan, China in late 2019, as of 15 March 2022, the COVID-19 pandemic has seen more than 450 million confirmed cases and 6 million confirmed deaths worldwide, causing unprecedented economic and social disruption.[1] Despite its global reach, it has become increasingly clear that the risks of initial infection, death and long-term consequences from COVID-19 are not distributed equally in society, including between different ethnic groups.[2]

In the early phase of the pandemic, initial studies sought to characterise factors associated with COVID-19 infection in patients admitted to hospital,[3 4] as only these patients received formal COVID-19 tests at the time. However, by focusing on this limited patient group, these studies had intrinsic biases which limited their generalisability to the wider population.[5]

To address these concerns, studies have sought to study COVID-19 infection risk in

larger population cohorts, such as the established UK Biobank project.[6] Early in the pandemic, COVID-19 test results for UK Biobank participants were linked to their biobank data, allowing participants' COVID-19 status to be followed.[7] These studies have identified several factors, including age, sex, comorbidities, ethnicity and socioeconomic status (SES) as significant determinants of COVID-19 infection, hospitalisation and mortality.[8–23] Specifically, studies have suggested that black and Asian people are at increased risk of testing positive, being hospitalised and dying from COVID-19, compared with white people.[9–12 16 18–21 23]

However, these studies were also prone to limitations. First, many gathered information from the UK Biobank only during the first wave of COVID-19 (February–May 2020), meaning there were relatively few positive cases to compare to the rest of the cohort, and wider population testing (so-called 'pillar 2' testing) only became available in the UK in early April 2020. Second, many only dealt with individual outcome measures—either testing positive, being hospitalised or death—meaning that the rest of the individual's journey through COVID-19 was not comprehensively assessed. This lack of detail may mask some differences between groups, thus limiting the implementation of findings concerning the role of ethnicity into public health policy in both the current and future pandemics. Furthermore, the lack of a multistate analysis could increase the chances of bias and erroneous conclusions, especially when focusing on ethnicity where inequities can lead to differential risks of transitions between testing, infection, hospitalisation and death.

This paper seeks to address several of these issues to delineate the influence of ethnicity on a variety of COVID-19 outcomes. We use UK Biobank data from the start of pillar 2 testing (April 2020) until September 2021, allowing an up-to-date comparison of a higher number of positive cases, across multiple waves of the pandemic. We analyse these data using a multistate model to be able to study participants' transitions between different 'COVID-19 states', given that the outcome of interest was just the first transition into any of the stages of a journey through COVID-19.

## METHODS
### Study design, setting and data sources
We conducted a cohort study using the UK Biobank cohort (UKB),[5] a large, non-commercial, long-term biobank project in the UK which stores medical data on around half a million people who were recruited between 2006 and 2010. All participants were aged between 40 and 69 years at the time of their registration with the project and will remain in follow-up for a minimum of 30 years. All participants provided demographic and lifestyle information, as well as blood, urine and saliva samples. COVID-19 testing data were provided by the UK Biobank through dynamic linkage with Public Health England's Second Generation Surveillance System.[7]

These data are only available for England and has 108 laboratories reporting positive tests, and 101 reporting negative tests too. Data on deaths were taken from the Office of National Statistics, which the UK Biobank updates regularly for their participants, and it is fully linked for England.[24]

We followed participants in this programme from 8 April 2020 to 30 September 2021, which is representative of the start of wider population testing (pillar 2) for COVID-19 in the UK to the latest available data entry point common to all different data sources (testing, hospitalisation, mortality records) at the time of analysis and writing. We limited our analysis to participants in England, given that England has a different multiple economic deprivation index to the other constituent countries of the UK, and testing data were only available for England.

Data from the UK Biobank were combined with National Health Service (NHS) Hospital Episode Statistics (HES) Admitted Patient Care data,[25] a programme which stores data on all hospital admissions, outpatient appointments and attendances, to build a comprehensive database that gathered information on COVID-19 testing, hospitalisation and mortality, as well as key variables including age, sex, ethnicity and SES for the entire cohort.

### Multistate framework
A multistate model[26] served as the framework for analysis, which has been used previously to describe risks of COVID-19 diagnosis, hospitalisation and death elsewhere.[27] These models allow the characterisation and analysis of individual transitions between different 'COVID-19 states'. In this study, all participants began as part of the UKB, who had never been tested for COVID-19. They could then progress to one of three separate states: testing negative, testing positive or direct hospitalisation (where testing positive for COVID-19 occurred on the same day as hospitalisation). Addition of a fifth health state of 'death' allowed for seven separate possible transitions: UKB to negative, UKB to positive, UKB to hospital, negative to positive, positive to hospital, positive to death (where hospitalisation occurred on the same day as death, or the patient was never hospitalised), and hospital to death (figure 1). Only the first negative test and subsequent positive test, or the first positive test, were counted, and each transition is unidirectional, so once an individual had a positive test, any subsequent negative tests are discounted.

### Participants and study size
We initially included all participants of the UK Biobank programme. We excluded participants if they live outside of England, had died prior to the start of follow-up (8 April 2020), had requested that their data not be used for research, or had either missing data on ethnic background or had answered 'do not know' or 'prefer not to say' to questions asking about ethnic background.

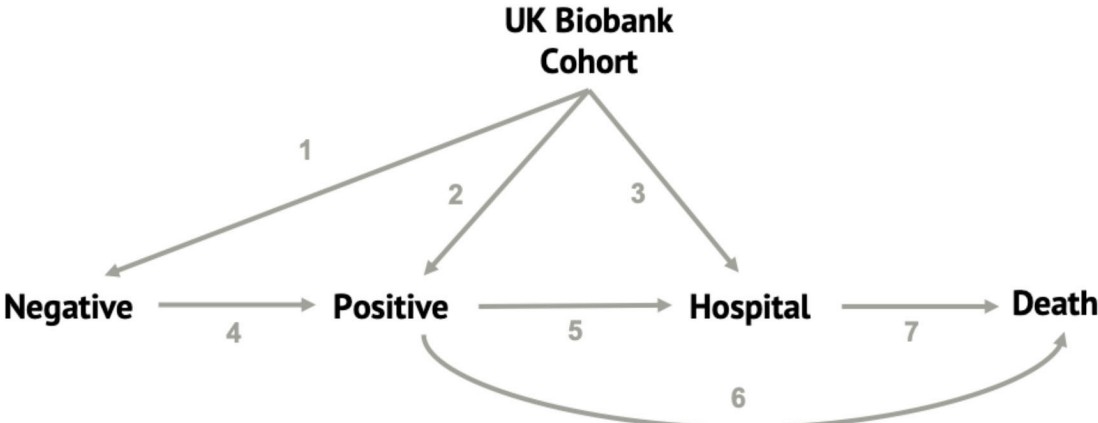

**Figure 1** Overview of the multistate model used in this study with the four 'COVID-19 states' and the possible transitions.

## Variables

The main 'exposure' of interest was ethnicity. Ethnicity was categorised into six groups, in line with the ethnic groupings specified by the UK Biobank at the time of recruitment to their programme. These are: Asian, black, Chinese, mixed, white and 'other'. Further detail on the ethnicities comprising each group can be found in the UKBiobank showcase (https://biobank.ndph.ox.ac.uk/showcase/field.cgi?id=21000). Language used in the reporting and discussion of ethnicity in the text is in line with published recommendations.[28] The main covariates for multivariable adjustment were sex, age and socioeconomic deprivation (Index of Multiple Deprivation quintiles) and comorbidities The Index of Multiple Deprivation combines information from seven unique domains of deprivation, to provide an overall relative measure of deprivation for 32844 neighbourhoods in England (lower layer super output areas).[29] Comorbidities were assessed using both hospitalisation data (HES) and primary care at the time of start of follow-up (8 April 2020). We used ICD-10 (International Classification of Diseases Revision 10) for HES,[30] and SNOMED and READ codes for primary care used previously,[31] using the categories of the Charlson Comorbidity Index.

## Outcomes

The outcomes of interest were results of COVID-19 tests, hospitalisation with COVID-19 and COVID-19-related death, gathered using the data sources discussed above. We deemed as COVID-19-related death or hospitalisation with COVID-19 those which occurred within 28 days of a positive SARS-CoV-2 test. COVID-19 was not required to be the primary diagnosis or reason for hospitalisation. Data on whether tests were positive or not was obtained from Public Health England as described elsewhere.[7]

Test count: Another outcome of interest was the test count, defined as the number of tests taken for each patient since the start of the follow-up period until the end of it.

Receiving a COVID-19 vaccine was defined as having a code READ code 'Y29e7' or SNOMED CT '1324681000000101' for the first dose and 'Y29e8' or '1324691000000104' for the second dose on the primary care data.

## Statistical methods

Baseline characteristics are presented as n (%) or median (IQR) overall and for each ethnicity category. Incidence of outcomes is shown as % of people who transition from one state to another during the whole study period. A Cox proportional hazards model was used to determine HRs for each group undergoing each transition in the multistate model, adjusted for the confounders outlined above. A 'clock forward' approach, modelling all transitions in the same timescale (time since entering the study), for time-to-events was used. The proportional hazards assumption was checked using the Kaplan-Meier estimator.[32]

Quasi-Poisson (to account for overdispersion) regressions were used to analyse the number of tests undertaken by each participant. It was further stratified by ethnic background.

For the analyses of vaccines, a Cox proportional hazards model was again used to determine HRs for each variable and transition: from unvaccinated to first dose and from first dose to second dose.

All statistical analyses were carried out using R: the tidyverse collection of packages for data curation,[33] the mstate and survival packages for multistate modelling and statistical regressions,[34 35] and kable and kableExtra for graphical representation.[36]

HRs for the positive to death transition are not shown, as most groups had no individuals undergoing this transition, causing CIs which were too wide to provide real information about relative risk.

## Patient and public involvement

No funding was available for patient or public involvement in this project. No patients were involved in setting the research question or the outcome measures. Patients were not invited to comment on the study design, not consulted to develop patient relevant outcomes or interpret the results, and not invited to contribute to the

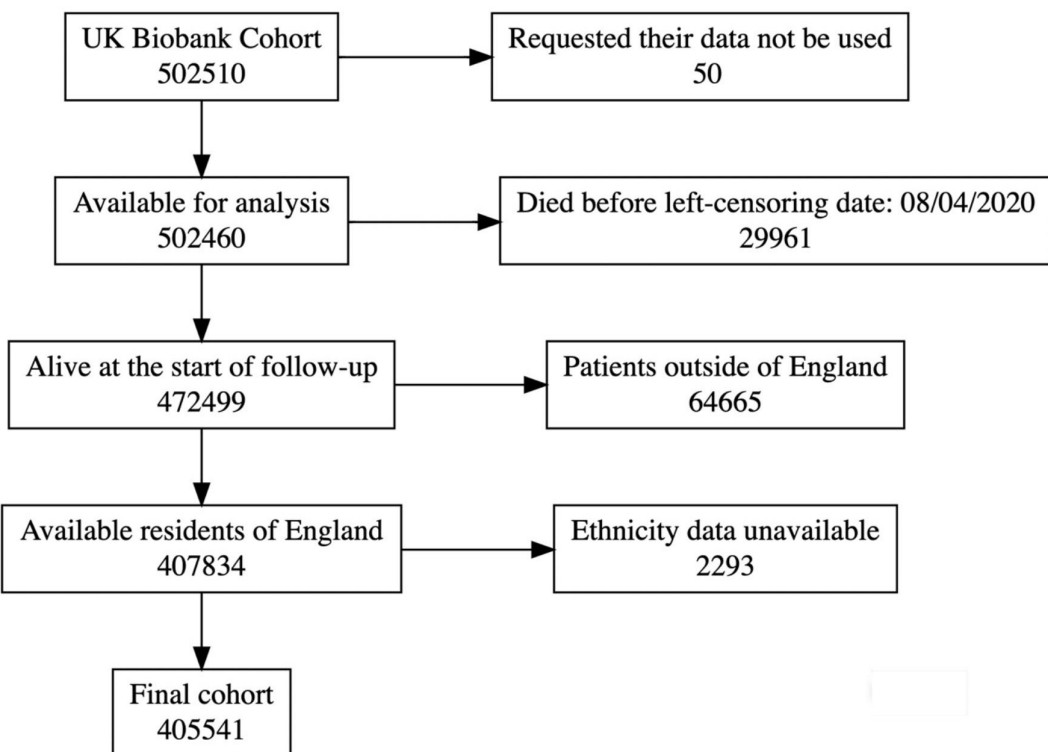

**Figure 2** Flow chart outlining stepwise exclusions of participants.

writing or editing of this document for readability or accuracy.

## RESULTS
### Baseline characteristics
After initial inclusion of all participants of the UK Biobank programme (n=502510), a total of 405541 (80.7%) participants were included in the final analysis (figure 2). The baseline characteristics of the final population are summarised in table 1. Our cohort was predominantly white with 381650 (94.1%), and the majority were female (224128–55.3%). The median age was 70.5 years, but the median age of white participants was around 5 years older than participants of other ethnic groups. Participants living in areas in the upper quintiles of socioeconomic status were more frequent in the cohort overall (29.7% in Q1 vs 12.8% in Q5), and there was a clear disparity in the distribution of participants between different quintiles based on ethnicity (online supplemental figure 1).

### Occurrence of COVID-19 outcomes
Of the entire cohort, 18172 (4.5%) individuals tested positive, and a further 3285 (0.8%) tested positive following a previous negative result. A total of 1490 individuals (6.9% of those who tested positive) were hospitalised following a positive result, and a further 2322 (0.6%) were directly hospitalised. Of those who were hospitalised, 662 individuals (17.4%) subsequently died within 28 days of a positive COVID-19 test. Full descriptive statistics of the transitions between each COVID-19 state are shown in

online supplemental table 1, stratified by ethnicity, socioeconomic deprivation, sex and age.

### Risks of testing positive, hospitalisation and death
Stratification of the cohort by ethnicity revealed differences between different ethnic groups in their risk of transitioning between different COVID-19 states (figure 3, online supplemental table 2). Compared with white participants, Asian participants had an increased risk of testing positive (1.40 (95% CI 1.30 to 1.51)) and direct hospitalisation (1.32 (95% CI 1.04 to 1.67)), and a reduced risk of testing negative (0.92 (95% CI 0.88 to 0.95)). Black participants had an increased risk of hospitalisation following a positive test (1.51 (95% CI 1.14 to 2.01)) and direct hospitalisation (1.54 (95% CI 1.22 to 1.94)). In contrast, Chinese participants had a reduced risk of testing negative (0.67 (95% CI 0.59 to 0.76)) and testing positive (0.41 (95% CI 0.29 to 0.58)). Participants with mixed or 'other' ethnicity did not have any notable differences in their risks of transitioning between any states compared with white participants. Once hospitalised, no ethnic group had increased risk of death compared with white participants.

Given the multistate model only accounts for the first positive or negative test an individual takes, Quasi-Poisson regression was used to compare differences in the amount of COVID-19 testing between groups (table 2). Black (1.15 (95% CI 1.08 to 1.22)) and 'other ethnicity' (1.13 (95% CI 1.03 to 1.23)) participants had an increased likelihood of testing for COVID-19, whereas in contrast Chinese

**Table 1** Cohort population baseline characteristics

| n | All | Asian | Black | Chinese | Mixed | Other | White |
|---|---|---|---|---|---|---|---|
| n | 405541 | 8791 | 7275 | 1324 | 2537 | 3964 | 381650 |
| Sex=male (%) | 181413 (44.7) | 4642 (52.8) | 3017 (41.5) | 468 (35.3) | 925 (36.5) | 1685 (42.5) | 170676 (44.7) |
| Age (median (IQR)) | 70.5 (62.8–75.8) | 65.2 (58.7–72.5) | 63 (57.9–70) | 65.1 (58.9–70.9) | 62.8 (57.6–70.4) | 64.6 (58.3–71.3) | 70.8 (63.2–76.1) |
| Socioeconomic deprivation | | | | | | | |
| Q1 (%) | 120443 (29.7) | 1345 (15.3) | 390 (5.4) | 370 (27.9) | 492 (19.4) | 567 (14.3) | 117279 (30.7) |
| Q2 (%) | 97002 (23.9) | 1452 (16.5) | 584 (8) | 253 (19.1) | 502 (19.8) | 559 (14.1) | 93652 (24.5) |
| Q3 (%) | 72708 (17.9) | 1825 (20.8) | 1106 (15.2) | 282 (21.3) | 436 (17.2) | 703 (17.7) | 68356 (17.9) |
| Q4 (%) | 63404 (15.6) | 2268 (25.8) | 2068 (28.4) | 227 (17.1) | 524 (20.7) | 962 (24.3) | 57355 (15) |
| Q5 (%) | 51984 (12.8) | 1901 (21.6) | 3127 (43) | 192 (14.5) | 583 (23) | 1173 (29.6) | 45008 (11.8) |
| Comorbidities | | | | | | | |
| Myocardial infarction (%) | 16389 (4) | 580 (6.6) | 152 (2.1) | 15 (1.1) | 75 (3) | 137 (3.5) | 15430 (4) |
| Congestive heart failure (%) | 11282 (2.8) | 267 (3) | 178 (2.4) | 7 (0.5) | 36 (1.4) | 89 (2.2) | 10705 (2.8) |
| Peripheral vascular disease (%) | 11643 (2.9) | 242 (2.8) | 140 (1.9) | 17 (1.3) | 54 (2.1) | 75 (1.9) | 11115 (2.9) |
| Cerebrovascular disease (%) | 18968 (4.7) | 347 (3.9) | 324 (4.5) | 26 (2) | 92 (3.6) | 157 (4) | 18022 (4.7) |
| Dementia (%) | 4299 (1.1) | 80 (0.9) | 87 (1.2) | 5 (0.4) | 16 (0.6) | 24 (0.6) | 4087 (1.1) |
| Chronic pulmonary disease (%) | 77899 (19.2) | 1761 (20) | 1315 (18.1) | 183 (13.8) | 536 (21.1) | 719 (18.1) | 73385 (19.2) |
| Rheumatological disease (%) | 15635 (3.9) | 373 (4.2) | 305 (4.2) | 24 (1.8) | 92 (3.6) | 128 (3.2) | 14713 (3.9) |
| Peptic ulcer disease (%) | 15830 (3.9) | 381 (4.3) | 334 (4.6) | 60 (4.5) | 94 (3.7) | 191 (4.8) | 14770 (3.9) |
| Mild liver disease (%) | 9890 (2.4) | 224 (2.5) | 200 (2.7) | 50 (3.8) | 69 (2.7) | 139 (3.5) | 9208 (2.4) |
| Diabetes (%) | 42222 (10.4) | 2687 (30.6) | 1693 (23.3) | 161 (12.2) | 301 (11.9) | 817 (20.6) | 36563 (9.6) |
| Hemiplegia (%) | 3278 (0.8) | 80 (0.9) | 95 (1.3) | 6 (0.5) | 20 (0.8) | 32 (0.8) | 3045 (0.8) |
| Moderate or severe renal disease (%) | 27065 (6.7) | 631 (7.2) | 750 (10.3) | 47 (3.5) | 155 (6.1) | 246 (6.2) | 25236 (6.6) |
| Cancer (%) | 55122 (13.6) | 665 (7.6) | 703 (9.7) | 106 (8) | 255 (10.1) | 365 (9.2) | 53028 (13.9) |
| AIDS (%) | 533 (0.1) | 9 (0.1) | 41 (0.6) | <5 (0.2) | 5 (0.2) | 17 (0.4) | 458 (0.1) |

Data are stratified by ethnicity and displays sex, age, socioeconomic status and comorbidities. Age is presented by group median age alongside IQR. All percentages (in brackets) are calculated with respect to group size n.

| | Hazard ratio | HR [95% CI] | | Hazard ratio | HR [95% CI] |
|---|---|---|---|---|---|
| **UKB-Negative** | | | **Negative-Positive** | | |
| Asian | 0.92 [0.88,0.96] | | Asian | 1.21 [0.99,1.48] | |
| Black | 1 [0.96,1.05] | | Black | 1.03 [0.83,1.29] | |
| Chinese | 0.67 [0.59,0.76] | | Chinese | 0.5 [0.19,1.33] | |
| Mixed | 1.05 [0.97,1.13] | | Mixed | 0.96 [0.65,1.42] | |
| Other | 1.04 [0.98,1.11] | | Other | 1.27 [0.96,1.67] | |
| **UKB-Positive** | | | **UKB-Hospital** | | |
| Asian | 1.4 [1.3,1.51] | | Asian | 1.32 [1.04,1.67] | |
| Black | 0.94 [0.85,1.03] | | Black | 1.54 [1.22,1.94] | |
| Chinese | 0.41 [0.29,0.58] | | Chinese | 1.32 [0.63,2.77] | |
| Mixed | 0.95 [0.81,1.12] | | Mixed | 0.87 [0.49,1.54] | |
| Other | 0.95 [0.83,1.09] | | Other | 1.22 [0.84,1.78] | |
| **Positive-Hospital** | | | **Hospital-Death** | | |
| Asian | 0.93 [0.72,1.19] | | Asian | 1.23 [0.81,1.85] | |
| Black | 1.51 [1.14,2.01] | | Black | 1.18 [0.74,1.88] | |
| Chinese | 1.78 [0.57,5.56] | | Chinese | 1.21 [0.3,4.87] | |
| Mixed | 1.22 [0.69,2.16] | | Mixed | 0.83 [0.31,2.24] | |
| Other | 0.94 [0.59,1.51] | | Other | 0.7 [0.26,1.89] | |

**Figure 3** Forest plots of ethnicity-stratified HRs, fully adjusted for age, sex and socioeconomic deprivation. The scale used for visualisation is linear but differs between each transition. The three dots reflect the x=1 axis. Fully adjusted for age, sex, socioeconomic deprivation and comorbidities.

participants had a reduced likelihood (0.70 (95% CI 0.58 to 0.86)).

In addition, we extracted data concerning COVID-19 vaccine uptake among our cohort, covering both the first and second doses of the vaccine (table 3). With regard to the first dose, all ethnic groups except Asian were less likely to have been vaccinated at the time of writing, compared with white participants. Asian and mixed ethnicity had similar chances than white participants to receive their second dose of the vaccine, whereas black and Chinese participants were less likely to.

## DISCUSSION
### Statement of principal findings
Our study of 405 541 participants of the UK Biobank demonstrates significant differences in risks of various COVID-19-related outcomes between different ethnic groups. Compared with white participants, we found that black and Asian participants have clearly increased risks of testing positive for COVID-19 and of being hospitalised for the disease. While our data suggest an increased risk of death following hospitalisation, the high uncertainty for this transition means that higher risk cannot be reliably asserted.

Both Asian and black participants had a higher likelihood of being tested for COVID-19, whereas Chinese participants were less likely to be, compared with white participants. In terms of vaccine uptake, our study shows white and Asian participants had a higher likelihood of

receiving their first vaccine dose compared with other ethnic groups. Compared with white participants, Asian and mixed ethnicity participants were more likely to receive their second vaccine dose, whereas black and Chinese participants were less likely to.

### Strengths and weaknesses of the study
Our study has several strengths. First, it includes a large number of participants from an established database, allowing extraction of both baseline characteristics and linked information related to COVID-19 testing and hospitalisation, helping to avoid issues around the self-reporting of COVID-19-related events and outcomes. Second, by choosing a start date for follow-up to coincide with the initiation of wider population testing in the community for COVID-19, we address issues of collider bias intrinsic to similar studies that focused on the early first wave of the pandemic, when testing was more limited. Third, our follow-up window lasts for 18 months, allowing the inclusion of higher numbers of COVID-19 cases, as well as providing an up-to-date analysis now that the UK has experienced multiple waves of the pandemic.

However, our study also has some weaknesses. Comparative studies have shown that the UK Biobank is not optimally representative of the wider population, with UK Biobank participants tending to be older, more female, healthier and less socioeconomically deprived, as we show in our description of our cohort's baseline characteristics, which limits the generalisability of our findings.[37] Our approach to confounders helps to address

**Table 2** Quasi-Poisson regression coefficients for number of tests taken, stratified by ethnicity and shown at different levels of confounding

| | Quasi-Poisson Risk ratio (95% CI) |
|---|---|
| Asian versus white | |
| Unadjusted | 1.10 (1.04 to 1.17) |
| Age and sex adjusted | 1.14 (1.07 to 1.22) |
| Fully adjusted* | 1.04 (0.97 to 1.10) |
| Black versus white | |
| Unadjusted | 1.27 (1.19 to 1.35) |
| Age and sex adjusted | 1.35 (1.26 to 1.44) |
| Fully adjusted* | 1.15 (1.08 to 1.22) |
| Chinese versus white | |
| Unadjusted | 0.64 (0.52 to 0.78) |
| Age and sex adjusted | 0.68 (0.55 to 0.83) |
| Fully adjusted* | 0.70 (0.58 to 0.86) |
| Mixed versus white | |
| Unadjusted | 0.98 (0.87 to 1.1) |
| Age and sex adjusted | 1.05 (0.93 to 1.19) |
| Fully adjusted* | 0.98 (0.87 to 1.10) |
| Other versus white | |
| Unadjusted | 1.15 (1.05 to 1.26) |
| Age and sex adjusted | 1.22 (1.11 to 1.33) |
| Fully adjusted* | 1.13 (1.03 to 1.23) |

Risk ratios are the exponential of the regression estimates, with the corresponding CIs. Quasi-Poisson model results are shown due to overdispersion in the sample.
*Fully adjusted for age, sex and socioeconomic deprivation.

**Table 3** Uptake risk of first and second doses of COVID-19 vaccines, stratified by ethnic background

| | First dose Risk ratio (95% CI) | Second dose Risk ratio (95% CI) |
|---|---|---|
| Asian versus white | | |
| Unadjusted | 0.82 (0.8 to 0.83) | 0.86 (0.84 to 0.88) |
| Age and sex adjusted | 0.99 (0.96 to 1.01) | 1.08 (1.05 to 1.10) |
| Fully adjusted* | 0.99 (0.97 to 1.01) | 1.09 (1.06 to 1.11) |
| Black versus white | | |
| Unadjusted | 0.52 (0.51 to 0.54) | 0.69 (0.67 to 0.71) |
| Age and sex adjusted | 0.61 (0.60 to 0.63) | 0.79 (0.77 to 0.81) |
| Fully adjusted* | 0.63 (0.61 to 0.64) | 0.81 (0.78 to 0.83) |
| Chinese versus white | | |
| Unadjusted | 0.65 (0.61 to 0.69) | 0.74 (0.69 to 0.78) |
| Age and sex adjusted | 0.78 (0.74 to 0.83) | 0.90 (0.85 to 0.95) |
| Fully adjusted* | 0.79 (0.74 to 0.83) | 0.90 (0.85 to 0.95) |
| Mixed versus white | | |
| Unadjusted | 0.66 (0.64 to 0.69) | 0.75 (0.72 to 0.79) |
| Age and sex adjusted | 0.85 (0.81 to 0.89) | 1.03 (0.99 to 1.08) |
| Fully adjusted* | 0.85 (0.82 to 0.89) | 1.04 (1.00 to 1.09) |
| Other versus white | | |
| Unadjusted | 0.62 (0.60 to 0.64) | 0.8 (0.77 to 0.82) |
| Age and sex adjusted | 0.74 (0.71 to 0.76) | 1.01 (0.97 to 1.04) |
| Fully adjusted* | 0.76 (0.73 to 0.79) | 1.02 (0.99 to 1.06) |

*Fully adjusted for age, sex, socioeconomic deprivation and comorbidities.

some of these issues, but we still have a very selected population. Although we adjust for socioeconomic status as a confounder, the deprivation index used is based on territorial units and not on individual characteristics, so it does not contain information pertaining to an individual. For example, we do not measure occupation, an important driver of COVID-19, and most of our population is in retirement age.

An important consideration is the potential confounding effect that differential vaccine uptake, illustrated in our results above, may have on our primary outcomes in the multistate model. Given the established effectiveness of vaccines in preventing severe disease, the vaccinated status of individuals in our study may have affected their risk of progressing through our multistate model. By design, our study could not include individuals who developed COVID-19 but did not test for it, or individuals who died with undiagnosed COVID-19. Also, the number of outcomes other than negative or positive tests in some ethnicity categories is quite low, so those should be interpreted with caution. We did not evaluate interactions that may be relevant, such as gender, age and SES with ethnicity, due to the low number of outcomes. We

also recognise that the UK Biobank's initial categorisation of ethnicity, used by necessity in this study, differs from the current UK Government Statistical Service ethnicity harmonised standard. Differences in linkage to tests, with some laboratories not reporting negative tests, and availability and differential use and reporting of home testing, could also have biased the results.

### Strengths and weaknesses in relation to other studies, discussing important differences in results

Our study has several strengths in comparison to previous studies examining COVID-19 outcomes using the UK Biobank. Our longer observation period and use of start dates in conjunction with wider population testing helps address intrinsic biases, and our multistate approach allows a more detailed assessment of different groups risks of specific outcomes than previously reported.[38]

This approach has allowed us to confirm several findings of previous studies, mainly conducted in a time frame where testing was only performed in hospital, namely that black and Asian individuals have a higher risk of testing positive for COVID-19 (indicating severe

disease as most were hospitalised).[2 11 23 39 40] We found that once more wide availability of tests started, we still find an increased risk of testing positive in hospital for black and Asian people, and increased risk of hospitalisation once testing positive for black participants, but a risk of testing positive outside hospital and testing negative similar to white participants. Our results also show increased testing rates among black and Asian participants also supports studies undertaken in the UK using other data sources and methods, such as large scale primary care data.[41]

[23 42] We found a test positivity rate similar to the general population but the case fatality rate seems to be higher than those of the general population for the same period. These differences are probably related to UK Biobank being composed of people of relatively older groups than the UK population.[42] We also found an increased mortality for those hospitalised with COVID-19 among black, Asian and Chinese ethnic groups, although with high uncertainty due to low numbers. This is in line with previous studies,[9 18] but probably with slightly lower due to those studies studying mortality in the general population, and our lethality among the hospitalised.

### Meaning of the study: possible explanations and implications for clinicians and policy-makers

The underlying causes of the inequalities demonstrated by our study are likely to be multifactorial, complex and varied.

Ethnic minority groups are disproportionately represented in lower socioeconomic groups, and deprivation is an established driver of poorer health outcomes. In our data, we could not disentangle deprivation effects from other related effects, such as occupation, other household inhabitants occupation, or living conditions. Data from the USA have suggested that ethnic minority groups are disproportionately represented in front-line roles which may increase exposure to the virus.[43] Similarly, in the UK, data from the Office for National Statistics have shown that 'key workers' are more likely than average to be from an ethnic minority group.[44] There is also evidence suggesting that ethnic minority groups in the UK are more likely to be employed in public-facing jobs where physical distancing is difficult (eg, transport).[40 45] Finally, there is evidence that ethnic minorities are more likely to live in overcrowded households, which represents a risk factor for COVID-19 diagnosis and severity.

In addition, structural and cultural racism and discrimination are also likely to be important contributors to the adverse outcomes experienced by ethnic minority groups during the pandemic.[46] It is established that ethnic disparities exist in a wide range of other healthcare contexts, such as maternal deaths[47] and sectioning under the Mental Health Act.[48] This has been thought to be multifactorial. Practical aspects such as transport availability and caring duties could have played a role.[49] Another possibly more important factor is racial discrimination in medical settings and other institutional barriers, such as inadequate communication, and culturally insensitive

care in the NHS. Studies have shown that implicit biases not only affect communication with patients, but also their experiences and outcomes of their clinical care.[50 51] This leads to lower trust in medical systems, and a higher susceptibility to targeted misinformation.[49 52 53] These inbuilt systemic biases towards ethnic minorities could be directly contributing to poorer outcomes after COVID-19 infection, due to delays in care, to the lower vaccination rate seen in the analyses, and to differential access to home testing and testing reporting.

### Unanswered questions and future research

Further research using the datasets from this study, for example, could investigate availability of testing and vaccination by territorial region, which is a source of systemic bias. More research into the underlying causes of ethnic disparities in pandemic-related outcomes will be important, especially when designing practical, effective and appropriate public health policy responses to future pandemics.

**Acknowledgements** This research has been conducted using the UK Biobank Resource under Application Number 46228. This work uses data provided by patients and collected by the NHS as part of their care and support. These data are copyrighted, 2022, NHS England. Reused with the permission of the NHS England and UK Biobank. All rights reserved.

**Contributors** DP-A and CP conceived and designed the study. TU, AP-U and MC acquired and performed analyses of the data, and all authors were involved in interpretation of the findings. FD, TU and AP-U drafted the work, and all authors were involved in critical revisions of the subsequent drafts. All authors gave final approval of the version to be published and agreed to be accountable for all aspects of the work. AP-U is responsible for the overall content as guarantor.

**Funding** The research was supported by the National Institute for Health Research (NIHR) Oxford Biomedical Research Centre (BRC). DP-A is funded through an NIHR Senior Research Fellowship (Grant number SRF-2018-11-ST2-004).

**Disclaimer** The views expressed in this publication are those of the author(s) and not necessarily those of the NHS, the National Institute for Health Research, or the Department of Health.

**Competing interests** DP-A's research group has received grant support from Amgen, Chesi-Taylor, Novartis and UCB Biopharma. His department has received advisory or consultancy fees from Amgen, Astellas, AstraZeneca, Johnson and Johnson, and UCB Biopharma and fees for speaker services from Amgen and UCB Biopharma. Janssen, on behalf of IMI-funded EHDEN and EMIF consortiums, and Synapse Management Partners have supported training programmes organised by DP-A's department and open for external participants organised by his department outside submitted work.

**Patient and public involvement** Patients and/or the public were not involved in the design, or conduct, or reporting, or dissemination plans of this research.

**Patient consent for publication** Not applicable.

**Ethics approval** This study involves human participants and the North West Multi-center Research Ethics Committee (MREC) approved the collection and use of UK Biobank data (16/NW/0274). UK Biobank approved the application used to access the data (Application 46228) on 17 July 2019 and its access to COVID-19 data on 17 April 2020. Participants gave informed consent to participate in the study before taking part.

**Provenance and peer review** Not commissioned; externally peer reviewed.

**Data availability statement** Data may be obtained from a third party and are not publicly available. Data are available after a request to UKBiobank.

**ORCID iDs**
Tomás Urdiales http://orcid.org/0009-0001-1478-5641
Francesco Dernie http://orcid.org/0000-0001-5344-1967
Albert Prats-Uribe http://orcid.org/0000-0003-1202-9153
Daniel Prieto-Alhambra http://orcid.org/0000-0002-3950-6346

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
