## [Reviewer comments · BMJ Open]

ARTICLE DETAILS

TITLE (PROVISIONAL)	The association between ethnic background and COVID-19 morbidity, mortality and vaccination in England: a multi-state cohort analysis using the UK Biobank
AUTHORS	Urdiales, Tomás; Dernie, Francesco; Català, Martí; Prats-Urbe, Albert; Prats, Clara; Prieto-Alhambra, Daniel

VERSION 1 – REVIEW

REVIEWER	Wif Miron, Rachel Gernter Institute for Epidemiology and Health Policy Research, Technology Assessment
REVIEW RETURNED	16-Jun-2022

GENERAL COMMENTS	General Comments The authors analyzed a unique, large, and comprehensive database, which combined several data sources, to allow a full picture of the COVID-19 burden and vaccine uptake. They wisely utilized diverse sophisticated statistical tools and methods to create a broad, well-adjusted analysis of the study outcomes. Title 1. Page 2, lines 3-4: The terms “status and health outcomes” are not the best terms to describe the current study. If I (the reviewer) had to dive into the Abstract to understand what the authors mean by “status”, then this is not a good title. Also, I feel that COVID-19 “status” is not the best description for the rate of confirmed cases, (I guess the term “health outcomes” relates to hospitalization or to hospitalization and death. In addition, most experts might not classify “vaccine uptake” as a health outcome. It is suggested to change the title, for example: The association between ethnic background and COVID-19 burden and vaccination: a multistate-cohort analysis Or The association between ethnic background and COVID-19 morbidity, mortality and vaccination: a multistate-cohort analysis Abstract 1. Participants:page 2, lines 41-2. Please add that only UK bank participants who live in England took part in the study. 2. Results: On page 2 lines 54 and lines 56, direct hospitalization is a term that at that point of the Abstract is not clear, since it is explained only in the Methods section of the main text on page 5, lines 34-36. Authors might add in line 51, “tested positive at the moment of hospital admission (i.e., “direct hospitalization”). 3. Conclusions page 3 lines 6-7: vaccine uptake is a better term than vaccine coverage Introduction
--

	1. Page 4, lines 10-11: “death and long-term consequences from COVID-19 are not distributed equally in society, especially between different ethnic groups”. Indeed, the focus of the manuscript is on ethnic disparities, however, the word “especially” is incorrect and might be replaced in a softer term, since socioeconomic gradient, in addition to ethnic differences, in COVID-19 burden as well as vaccination has been demonstrated in many countries, for example, see Luxenburg et al. Sociodemographic and ethnic disparities in COVID-19 burden: Changing patterns over four pandemic waves in Israel. Journal of Epidemiology & Community Health. 2. Page 4, lines 13-17. This paragraph needs re-writing. Firstly, many studies analyzed sociodemographic characteristics among the general (not just the hospitalized) population, see for example Saban et al: Covid-19 morbidity in an ethnic minority: changes during the first year of the pandemic. Public Health. 2021 Sep;198: 238-244. Secondly, the Collider Bias of non-representative sampling, mentioned in reference 4, is not clear in this context. To the best of my understanding, the authors mean that sampling among hospitalized patients is not representative of the entire population. It is not clear, therefore, why write “largely due to limited testing capabilities at the time” and how is this fact related to the sampling of hospitalized people only. Also, limited testing capacity in the first stages of the pandemic, as described in the UK, does not necessarily represent other countries. Please clarify your meaning, relating to reference 4. 3. Page 4, line 38. Lack of granular (translation = coarse, rough) detail – the use of “granular” is not clear in this sentence. It might be replaced. Methods 1. Variables, page 6 lines 7: Index of Multiple Deprivation: Please add a short explanation and reference, since this is a widely used measure in the UK but less recognized outside the UK. Results: 1. Page 7, lines 12-14: Please indicate if the quantiles of the Index of Multiple Deprivation were defined by an identical (one-fifth) number of citizens in each quantile, otherwise the statement that “the upper quantiles of socio-economic status were over-represented” might be inaccurate. Not every national measurement system divides the quintiles or quartiles into an identical or almost identical numbers of individuals in each category. 2. Page 8, lines 8-9:....”(17.4%) subsequently died” – please add subsequently died within 28 days of the positive COVID-19 test; indeed this is stated in the methods section, however, it is important to bear it in mind here too. 3. Page 8, lines 14-25: This paragraph bears the title “Risks of testing positive hospitalization and death”, however, death is not discussed in this paragraph at all, despite interesting findings presented in detail in Supplementary Table 2. Indeed, the authors claim that “stratification of the cohort by ethnicity revealed differences between different ethnic groups in their risk of transitioning between different COVID states (and understandably, death is included as the last link in the chain of events), however, all other transitions are detailed while the hospital-death link is missing. Discussion A general comment:
--	--

	1. It is suggested to elaborate more deeply on the meaning of the important findings of this study, with special emphasis on the mismatch between COVID-19 burden and vaccine uptake. One would wish that populations like the Black minority group, who demonstrated higher rates of testing positive, hospitalization and death, would be willing to receive the COVID vaccine, but this is not the case, unfortunately. 2. Page 11 line 9 – clearly increased risks (not “clear”) 3. Page 12, line 3. Increased testing rates (instead of testing prevalence; I feel less comfortable using “prevalence” for “testing”. 4. Page 12, lines 16-21 contain important and well-stated limitations of the study and can be moved to the limitation section. Limitations: 1. The authors might consider adding a limitation: They write on page 6 lines 45-47, “Hazard ratios for the Positive to Death transition are not shown, as most groups had no individuals undergoing this transition”.....The meaning of very few individuals not going through the hospital before deterioration to death might be related to cases of people refusing hospitalization, not arriving on time to the hospital, and rapidly deteriorating in the community (whether at their homes or at nursing homes). This group might be over-represented by more deprived, ethnic minority residents. 2. Please add to the limitation the fact mentioned in the “meaning of the study section” on using area-level, instead of individual-level, deprivation data. Figure 1, Figure 2, and Figure 3 (pages 18-20) do not have a title indicating which figures are presented there, although one can guess it from the order following the figure legends.
--	--

REVIEWER	Hua, Miao Jenny Northwestern University
REVIEW RETURNED	14-Jul-2022

GENERAL COMMENTS	The articles tests hypotheses regarding the associations between ethnicity and various COVID-related outcomes. Its strength is in examining multiple outcomes including testing, test positivity, hospitalization, death and vaccine receipt in the same cohort of patients from the UK Biobank through multi-state modeling. The model includes five states excluding vaccination and seven transitions through the five states. However, several issues/questions may need to be addressed prior to the decision to publish. 1) Race/ethnic categorization. The six ethnic groups considered (White, Black, Chinese, Asian, Mixed, Other) are difficult to parse and compare. In common parlance at least in the US (and in ref 26, the Flanagan et al. article from JAMA that the authors cite), “Chinese” might be the only ethnicity while the other groups are generalizations of external characteristics (in the US, they would termed “race,” although race and ethnicity are often used interchangeably, perhaps especially so in the UK). It is also unclear whether the category “Asian” includes Chinese – I would assume no, but what ethnic categories are included? Clarifications of what specific ethnicities are encompassed, especially under “Asian” and “Other,” would make the results more interpretable. 2) Why left censor at April 8, 2020 rather than adjust for different phases of the pandemic or subject the model to sensitivity analysis? The authors explain that they chose to focus on outcomes post-pillar 2 testing, but differential accessibility of
--

	testing over different phases of the pandemic is an important confounder of ethnic differences in testing and test positivity. The well-known differences in outcome severity over different phases of the pandemic should also be accounted for. 3) Why were no adjustments made for comorbid conditions? If this data is available through the Biobank, standard adjustments for baseline health status should be done before differential rates of hospitalization and death can be attributed to ethnic differences. If it was not available, this should be described as a limitation. 4) How did differential vaccination status interact with ethnicity-related risks for test positivity and hospitalization? The authors examine vaccine uptake as an outcome but it is also an important confounder for ethnic differences in test positivity, hospitalization and death. 5) Interpretation within existing literature lacking. A great deal has been published on the associations between race/ethnicity and COVID outcomes, much using the UK Biobank. The authors recognize this and cite a number of them on p. 4 but there is minimal situating of the results within the wider literature. Are they consistent with prior results or do they uncover something previously overlooked? 6) Overall, interpretability of the results can be improved. The authors attribute ethnic differences in outcomes to socioeconomic differences, even though these adjustments did not significantly alter hazard ratios in their model. They also attribute differential outcomes to structural racism without positing specific mechanisms by which racism interacts with COVID outcomes.
--	--

REVIEWER	Fung, Christopher University of Michigan
REVIEW RETURNED	18-Jul-2022

GENERAL COMMENTS	Thank you for the opportunity to review the manuscript entitled “The association between ethnic background and COVID-19 testing, status and health outcomes: a multi-state cohort analysis”. This is a large population-based cohort study leveraging the UK Biobank’s half a million participants. The study authors utilized this existing longitudinal study to follow participants and track their time of transition between various Covid health states in a multistate model of Covid-19 testing and infection. Participants were followed from April 2020 to September 2021 which is a period that roughly includes the first major surge of Covid-19 infections in the UK at the end of 2020 and the beginning of another major surge of infections at the end of 2021. The goal of this study was to determine the association between ethnic background and various states (receiving first test, testing positive, hospitalization, vaccination and death) of health related to Covid. The authors report a primary finding that Black and Asian participants were more likely to be tested for Covid, more likely to test positive and more likely to be hospitalized. White and Asian participants were more likely to be vaccinated. My overall recommendation is to accept this study for publication with minor revisions. While this study is one of many studies that report on ethnic and racial differences in health during the pandemic, the study leverages a very large cohort with reliable follow up and linkage to other databases. Thus, this is an important confirmation of other reports of ethnic and racial disparities in outcomes related to Covid-19. My primary recommendation to the authors is to revise the framing of the manuscript to emphasize the likely sample bias in the UK Biobank
--

given is ~95% White participants and proportionally small sampling of Asian, Chinese, Black and other participants. The authors report that minority participants in this study are likely to have similar characteristics as minority populations in the UK but their participation in the Biobank likely makes the participants different than the population (i.e. non-White participants in the study are probably more different than the population than their White counterparts). While the multivariable models do adjust for sex, gender and SES based on territorial data as a proxy for individual data, one of the strengths of the UK Biobank is robust individual data on demographics, medical history and linkage to outside databases. Its not clear to me why this individual level data was not used as adjustment for comorbidities, individual occupation, SES, etc. as it should be possible given available linkages/data. An explanation of why this was not done would be sufficient.

Detailed Comments:

RECORD statement – Appropriate for this study design and thank you for including this checklist.

Abstract – The abstract is a concise description of the study, underlying population, outcomes and results. If space allows, I recommend a little more description of the analysis as its not clear from the abstract what type of analysis was done if the reader is not familiar with multistate survival analyses (not a technique that I personally use). Perhaps a line about modeling time-to transition states would be helpful. Also, for participants, its probably worth noting just how small all the non-white part of the cohort was. It would be informative to have a line like “405,41 total subjects were eligible for analysis and 6% were non-white”

Introduction:

Page 4, Line 8 – stats for covid cases probably need a reference even though they are so ubiquitous.

Methods:

Page 5, Line 8 – would be useful to provide some information from reference about quality of linkage between biobank and testing/vaccine/outcomes data. From the reference (#6), it seems like there is incomplete reporting of negative tests from some labs, patients that may not link correctly and temporal differences in availability of testing over the study period that may bias the results.

Page 5 Line 22 – same as above. A top line reporting of quality of linkage/missing data would be helpful.

Page 5, Line 44-45 – not sure that the approach of unidirectional state transitions gives a “more detailed” overview but this is nitpicky. Consider reframing to simplification of the model or emphasizing that the outcome of interest was just the first transition into any of these states.

Page 5, Line 52-53 – missing exposure data is presented in the study flow diagram. What about missing data from the outcomes (patients that did not link into other databases).

Page 6, Line 6-7 – covariates in the final models are listed but prefer if this paragraph would have some justification for inclusion of each variable. Could be done with references or expert opinion as the included covariates are the usual ones. However, some explanation of territorial measure of socioeconomic deprivation versus individual data should be detailed here. Its not clear why this measure was used instead of an individual measure (was it just not available in the existing data or linked databases?). Also, given the regional variability of surges in Covid-19, was there consideration of a model that includes a variable to capture regional prevalence (within England) as this may greatly influence all outcomes. Was there consideration of other variables such as comorbidities, primary language other than English, immigrant status or other variables known to influence healthcare utilization by ethnicity or race.

Page 6, Line 11-22 – This section should be expanded significantly with definitions of each outcome explicitly stated. Which Covid tests counted as positive/negative tests. Report LOINCs. Please make it clear how hospitalization “with covid” was defined. Were only hospitalizations where Covid was the primary diagnosis used or were all hospitalizations used as the outcome? Same with 28 day mortality – please clarify whether or not Covid was required to be primary diagnosis or reason for hospitalization.

Page 6, Line 30-34 – would be helpful (maybe in supplemental) to provide the exact model specifications for each transition modeled as it seems like there will be both left and right censoring, maybe a supplemental table of all the models, censoring, and maybe some notes about fit testing. Its not needed in the main text but would be informative as supplemental. Could consider reporting bivariate associations here too such as median time to (transition state from prior) with Cis.

Page 6, Lines 35-36 – this appears to be an analysis of an outcome, number of tests, that is not described above in the outcomes section. Please report how this was defined above.

Page 6, Statistical methods section – please report how model assumptions were tested and if any other model diagnostics were performed. Were any interactions assessed? I recommend doing this as a sensitivity analysis because in other health disparities, gender-race interactions or race-age interactions are common. There could be others. If no interactions were evaluated that’s ok, the group sizes are very large but should be reported as a limitation.

Results:

Page 7, Line 15 – the outcome counts for the non-white groups are very low (single digits) so its difficult to draw any real conclusions from these groups and should be discussed as a limitation later in discussion.

Page 8, Line 5-11 – 4.5% test positivity rate seems about right relative to general UK population at 18 months into the pandemic. However, death rate (~3% with 662/18,172) seems high suggesting overrepresentation in the numerator. Please explain in the discussion.

	Page 8, line 16 – in figure 3, the forest plot, please include an X axis. Its difficult to tell if its linear or log scale and this would inform the take away message from the size of the error bars. Discussion: Page 11, Line 37-41 – Discussion of weaknesses should include interpretation of how differences between Biobank and general UK population may affect generalizability of results. Also, do these differences (biobank vs general population) occur in ethnic/racial minorities? Some discussion of this would inform the reader’s interpretation of results as well. For example, are higher socioeconomic status Asian’s overrepresented relative to the general population of Asians in the UK? Page 11, Line 41-44 – Please address how increasing availability of home tests during the study period influences the results. For example, do white participants have a lower risk of testing positive because they are home testing and not reporting the result? Page 12, Line 12-28 – The authors acknowledge that the reasons for racial and ethnic disparities in Covid outcomes are complex yet speculate about shift work as a specific reason for the observed disparities in this study. To me this seems out of place and a broader summary existing evidence about covid outcomes disparities would be more appropriate. The manuscript does not discuss or present data regarding shift work or key workers at all and only makes the connection in the discussion. Page 12, Line 29-37 and 43-46j – This discussion of systemic biases in health care is pertinent and perhaps highlights potential next steps in this work? For example, it seems possible from the existing datasets used in this study to investigate availability of testing and vaccination by territorial region which is a testable and intervenable source of systemic bias. Would love to see proposed future analyses such as this in this section.
--	---

VERSION 1 – AUTHOR RESPONSE

Reviewer: 1

Dr. Rachel Wilf Miron, Gernter Institute for Epidemiology and Health Policy Research, Tel Aviv University

Comments to the Author:

General Comments

The authors analyzed a unique, large, and comprehensive database, which combined several data sources, to allow a full picture of the COVID-19 burden and vaccine uptake. They wisely utilized diverse sophisticated statistical tools and methods to create a broad, well-adjusted analysis of the study outcomes.

Thank you for your comments, they have greatly helped improve the manuscript and made it more accessible.

Title

1. Page 2, lines 3-4: The terms “status and health outcomes” are not the best terms to describe the current study. If I (the reviewer) had to dive into the Abstract to understand what the authors mean by “status”, then this is not a good title. Also, I feel that COVID-19 “status” is not the best description for the rate of confirmed cases, (I guess the term “health outcomes” relates to hospitalization or to hospitalization and death. In addition, most experts might not classify “vaccine uptake” as a health outcome. It is suggested to change the title, for example:

The association between ethnic background and COVID-19 burden and vaccination: a multistate-cohort analysis

Or

The association between ethnic background and COVID-19 morbidity, mortality and vaccination: a multistate-cohort analysis

Thank you for your suggestion. The title is much more informative now: “The association between ethnic background and COVID-19 morbidity, mortality and vaccination in England: a multi-state cohort analysis using the UK Biobank”

Abstract

1. **Participants:page 2, lines 41-2. Please add that only UK bank participants who live in England took part in the study.**

Thank you, we have added your suggestion.

2. **Results: On page 2 lines 54 and lines 56, direct hospitalization is a term that at that point of the Abstract is not clear, since it is explained only in the Methods section of the main text on page 5, lines 34-36. Authors might add in line 51, “tested positive at the moment of hospital admission (i.e., “direct hospitalization”).**

We have incorporated it into our manuscript. Thank you for your valuable input.

3. **Conclusions page 3 lines 6-7: vaccine uptake is a better term than vaccine coverage**

Thank you, we have incorporated the change.

Introduction

1. **Page 4, lines 10-11: “death and long-term consequences from COVID-19 are not distributed equally in society, especially between different ethnic groups”. Indeed, the focus of the manuscript is on ethnic disparities, however, the word “especially “is incorrect and might be replaced in a softer term, since socioeconomic gradient, in addition to ethnic differences, in COVID-19 burden as well as vaccination has been demonstrated in many countries, for example, see Luxenburg et al. Sociodemographic and ethnic disparities in COVID-19 burden: Changing patterns over four pandemic waves in Israel. Journal of Epidemiology & Community Health.**

Thank you, we have changed it to ‘including’ to increase clarity.

2. **Page 4, lines 13-17. This paragraph needs re-writing. Firstly, many studies analyzed sociodemographic characteristics among the general (not just the hospitalized) population, see for example Saban et al: Covid-19 morbidity in an ethnic minority: changes during the first year of the pandemic. Public Health. 2021 Sep;198: 238-244.**

Secondly, the Collider Bias of non-representative sampling, mentioned in reference 4, is not clear in this context. To the best of my understanding, the authors mean that sampling among hospitalized patients is not representative of the entire population. It is not clear, therefore, why write “largely due to limited testing capabilities at the time” and how is this fact related to the sampling of hospitalized people only. Also, limited testing capacity in the first stages of the pandemic, as described in the UK, does not necessarily represent other countries. Please clarify your meaning, relating to reference 4.

The two studies we cite (2 and 3) in the first sentence of this paragraph relate to very early studies conducted in the first phase of the pandemic, specifically in the UK, which is the same setting as our own paper, and thus were felt to be most relevant when setting out the context of our paper.

The interpretation of the next sentence is correct. Limited testing capabilities (for the general public) early in the pandemic meant early studies focussed only on hospitalised patients (who had therefore received a COVID test). This was an important consideration within the UK when interpreting these studies. It also directly relates to the design of our study which commences once widespread testing was available. We acknowledge that every country had different responses and timelines during the pandemic, however we feel that it was important to describe the context and background most relevant to the setting of our own study.

We have changed the wording of this paragraph to make this clearer. Reference 4 explains issues of collider bias.

3. Page 4, line 38. Lack of granular (translation = coarse, rough) detail – the use of “granular” is not clear in this sentence. It might be replaced.

We thank the reviewer for their comment. We understand that this word is polysemic with contradictory meanings and could lead to misinterpretation, so we have changed it for “detailed”.

Methods

1. Variables, page 6 lines 7: Index of Multiple Deprivation: Please add a short explanation and reference, since this is a widely used measure in the UK but less recognized outside the UK.

Thank you for this comment. We have added a short explanation and a reference: “The Index of Multiple Deprivation combines information from seven unique domains of deprivation, to provide an overall relative measure of deprivation for 32,844 neighbourhoods in England (Lower-layer Super Output Areas). “

Results:

1. Page 7, lines 12-14: Please indicate if the quantiles of the Index of Multiple Deprivation were defined by an identical (one-fifth) number of citizens in each quantile, otherwise the statement that “the upper quantiles of socio-economic status were over-represented” might be inaccurate. Not every national measurement system divides the quintiles or quartiles into an identical or almost identical numbers of individuals in each category.

We thank the reviewer for their comment. The Index of Multiple Deprivation quantiles were indeed defined by identical numbers of LSOA areas, a grouping based on adjacent zip codes. There is not the same number of people in each quintile, so the misrepresentation of UKBiobank population

structure is even higher: In the most affluent quintile there is 29% of UKB population, and in England it only represents around 7% of the population.

2. Page 8, lines 8-9:....”(17.4%) subsequently died” – please add subsequently died within 28 days of the positive COVID-19 test; indeed this is stated in the methods section, however, it is important to bear it in mind here too.

Thank you, added.

3. Page 8, lines 14-25: This paragraph bears the title “Risks of testing positive hospitalization and death”, however, death is not discussed in this paragraph at all, despite interesting findings presented in detail in Supplementary Table 2. Indeed, the authors claim that “stratification of the cohort by ethnicity revealed differences between different ethnic groups in their risk of transitioning between different COVID states (and understandably, death is included as the last link in the chain of events), however, all other transitions are detailed while the hospital-death link is missing.

We thank the reviewer for their comment. We have now added a sentence summarising the death aspect of our results to the paragraph.

Discussion

A general comment:

1. It is suggested to elaborate more deeply on the meaning of the important findings of this study, with special emphasis on the mismatch between COVID-19 burden and vaccine uptake. One would wish that populations like the Black minority group, who demonstrated higher rates of testing positive, hospitalization and death, would be willing to receive the COVID vaccine, but this is not the case, unfortunately.

This is a very important point. We were not able to draw conclusions differentially the time when the vaccine was available and the time when it wasn't in terms of outcomes due to low power, but, although not presented due to huge confidence intervals, the same ethnic groups as shown in the analyses were already having worse outcomes before the introduction of the vaccine. We have extended the discussion about structural racism and how mistrust in the healthcare systems, united to other factors, could explain worse outcomes and lower vaccination.

2. Page 11 line 9 – clearly increased risks (not “clear”)

Thank you for your feedback. We have incorporated it!

3. Page 12, line 3. Increased testing rates (instead of testing prevalence; I feel less comfortable using “prevalence” for “testing”).

Incorporated your feedback. Thank you.

4. Page 12, lines 16-21 contain important and well-stated limitations of the study and can be moved to the limitation section.

We have added a comment on this fact in the limitations section, and added a shorter discussion of it where it was originally to add context to the following paragraphs.

Limitations:

1. The authors might consider adding a limitation: They write on page 6 lines 45-47, “Hazard ratios for the Positive to Death transition are not shown, as most groups had no individuals undergoing this transition”.....The meaning of very few individuals not going through the hospital before deterioration to death might be related to cases of people refusing hospitalization, not arriving on time to the hospital, and rapidly deteriorating in the community (whether at their homes or at nursing homes). This group might be over-represented by more deprived, ethnic minority residents.

We thank the reviewer for their comment. This is a relevant point, which we feel is addressed in the Meaning of Study section of the manuscript. Nevertheless, the data does not show indications of either presence or absence of such a phenomenon for the Positive to Death transition. In the entire dataset there are only 129 occurrences, 120 of which are of White participants. With such small numbers it is not possible to perform reliable comparative analysis, although for those ethnic minority groups where there are any occurrences of this transition, the frequency (in relation to group size) is nearly identical to that of Whites.

2. Please add to the limitation the fact mentioned in the “meaning of the study section” on using area-level, instead of individual-level, deprivation data.

We have added a comment on this fact in the limitations section too.

Figure 1, Figure 2, and Figure 3 (pages 18-20) do not have a title indicating which figures are presented there, although one can guess it from the order following the figure legends.

Seems like an automatic formatting error from the pdf conversion. We have added the titles to the end of the manuscript for reference.

Reviewer: 2

Dr. Miao Jenny Hua, Northwestern University

Comments to the Author:

The article tests hypotheses regarding the associations between ethnicity and various COVID-related outcomes. Its strength is in examining multiple outcomes including testing, test positivity, hospitalization, death and vaccine receipt in the same cohort of patients from the UK Biobank through multi-state modeling. The model includes five states excluding vaccination and seven transitions through the five states. However, several issues/questions may need to be addressed prior to the decision to publish.

1) Race/ethnic categorization. The six ethnic groups considered (White, Black, Chinese, Asian, Mixed, Other) are difficult to parse and compare. In common parlance at least in the US (and in ref 26, the Flanagan et al. article from JAMA that the authors cite), “Chinese” might be the only ethnicity while the other groups are generalizations of external characteristics (in the US, they would be termed “race,” although race and ethnicity are often used interchangeably, perhaps especially so in the UK). It is also unclear whether the category “Asian” includes Chinese – I would assume no, but what ethnic categories are included? Clarifications of what specific ethnicities are encompassed, especially under “Asian” and “Other,” would make the results more interpretable.

We thank the reviewer for their comment. While Flanagan et al. informs the writing standards for discussions of ethnicity in this manuscript, the specific categorisations adopted are necessarily those available in the UK Biobank dataset, which is based on the government standard at the time of registration. Clarifications on what these ethnic categories encompass and how they are defined is available in the UK Biobank showcase (<https://biobank.ndph.ox.ac.uk/showcase/field.cgi?id=21000>). We have added this link to the methods section discussing the ethnicity categories.

2) Why left censor at April 8, 2020 rather than adjust for different phases of the pandemic or subject the model to sensitivity analysis? The authors explain that they chose to focus on outcomes post-pillar 2 testing, but differential accessibility of testing over different phases of the pandemic is an important confounder of ethnic differences in testing and test positivity. The well-known differences in outcome severity over different phases of the pandemic should also be accounted for.

We thank the reviewer for their comment. It is certainly true that differential accessibility of testing and hospital access may have been an important confounder when it comes to ethnic differences, and an analysis of this kind was attempted during the early stages of this study (according to the different phases or ‘waves’ of the pandemic). Unfortunately, once the available cohort is broken down for statistical analysis according to all relevant individual characteristics and further stratified into the different states of the multi-state model, there aren’t enough occurrences across all states and profiles to then separate into different phases of the pandemic and obtain reliable results. This could be performed for comparisons of age, sex or deprivation index (as these groupings are much larger), but there isn’t enough data to do so for ethnic differences. Instead, the study analyses aggregated differences throughout the pandemic starting with the introduction of pillar 2 testing on April 8th.

3) Why were no adjustments made for comorbid conditions? If this data is available through the Biobank, standard adjustments for baseline health status should be done before differential rates of hospitalization and death can be attributed to ethnic differences. If it was not available, this should be described as a limitation.

We thank the reviewer for their comment. We have now added adjustments for comorbidities, and these are now presented in the manuscript. We have added a brief explanation in methods, a Table 2 showing the comorbidities at baseline, and we have updated the results with the further adjustment.

4) How did differential vaccination status interact with ethnicity-related risks for test positivity and hospitalization? The authors examine vaccine uptake as an outcome but it is also an important confounder for ethnic differences in test positivity, hospitalization and death.

We thank the reviewer for raising this important point. We have now added a section to our limitations section of our discussion to highlight this consideration.

5) Interpretation within existing literature lacking. A great deal has been published on the associations between race/ethnicity and COVID outcomes, much using the UK Biobank. The authors recognize this and cite a number of them on p. 4 but there is minimal situating of the results within the wider literature. Are they consistent with prior results or do they uncover something previously overlooked?

We have greatly expanded on the interpretation with existing literature and added more references.

6) Overall, interpretability of the results can be improved. The authors attribute ethnic differences in outcomes to socioeconomic differences, even though these adjustments did not significantly alter hazard ratios in their model. They also attribute differential outcomes to structural racism without positing specific mechanisms by which racism interacts with COVID outcomes.

We thank the reviewer for their comment. The principal objective of the study is to measure and describe differences present in the data; accurately attributing causality for these observations is outside of its scope. In this context, it is discussed under *Meaning of the study* that the deprivation index employed may not completely reflect socio-economic status, because it is based on larger territorial units (LSOA areas) and not on individual/household characteristics. It is possible that this lack of granularity in the measurement of socio-economic status (not accounting for participants' occupations) partly explains the large differences observed across ethnic groups. This is brought up as an example of possible residual confounding not accounted for (and thus a possible explanation of the results), but by no means does the study attribute to it the ethnic differences observed. We have also expanded the discussion about structural racism and its potential impact on all covid outcomes and vaccination.

Reviewer: 3

Dr. Christopher Fung, University of Michigan

Comments to the Author:

Dear Editorial Team,

Thank you for the opportunity to review the manuscript entitled “The association between ethnic background and COVID-19 testing, status and health outcomes: a multi-state cohort analysis”. This is a large population-based cohort study leveraging the UK Biobank’s half a million participants. The study authors utilized this existing longitudinal study to follow participants and track their time of transition between various Covid health states in a multistate model of Covid-19 testing and infection. Participants were followed from April 2020 to September 2021 which is a period that roughly includes the first major surge of Covid-19 infections in the UK at the end of 2020 and the beginning of another major surge of infections at the end of 2021. The goal of this study was to determine the association between ethnic background and various states (receiving first test, testing positive, hospitalization, vaccination and death) of health related to Covid. The authors report a primary finding that Black and Asian participants were more likely to be tested for Covid, more likely to test positive and more likely to be hospitalized. White and Asian participants were more likely to be vaccinated.

My overall recommendation is to accept this study for publication with minor revisions. While this study is one of many studies that report on ethnic and racial differences in health during the pandemic, the study leverages a very large cohort with reliable follow up and linkage to other databases. Thus, this is an important confirmation of other reports of ethnic and racial disparities in outcomes related to Covid-19. My primary recommendation to the authors is to revise the framing of the manuscript to emphasize the likely sample bias in the UK Biobank given is ~95% White participants and proportionally small sampling of Asian, Chinese, Black and other participants. The authors report that minority participants in this study are likely to have similar characteristics as minority populations in the UK but their participation in the Biobank likely makes the participants different than the population (i.e. non-White participants in the study are probably more different than the population than their White counterparts). While the multivariable models do adjust for sex, gender and SES based on territorial data as a proxy for individual data, one of the strengths of the UK Biobank is robust individual data on demographics, medical history and linkage to outside databases. Its not clear to me why this individual level data was not used as adjustment for comorbidities, individual occupation, SES, etc. as it should be possible given available linkages/data. An explanation of why this was not done would be sufficient.

Thank you for your comments, that have helped to improve the paper. We have worked to add comorbidities at an individual level to the paper, that confirm the results and have made the paper more robust.

Detailed Comments:

RECORD statement – Appropriate for this study design and thank you for including this checklist.

Abstract – The abstract is a concise description of the study, underlying population, outcomes and results. If space allows, I recommend a little more description of the analysis as its not clear from the abstract what type of analysis was done if the reader is not familiar with multistate survival analyses (not a technique that I personally use). Perhaps a line about modeling time-to transition states would be helpful. Also, for participants, its probably worth noting just how small all the non-white part of the cohort was. It would be informative to have a line like “405,41 total subjects were eligible for analysis and 6% were non-white’

Thank you for this comment. We have added a line stating that 6% were non-white to the abstract.

Introduction:

Page 4, Line 8 – stats for covid cases probably need a reference even though they are so ubiquitous.

Thank you for this, we have added a reference.

Methods:

Page 5, Line 8 – would be useful to provide some information from reference about quality of linkage between biobank and testing/vaccine/outcomes data. From the reference (#6), it seems like there is incomplete reporting of negative tests from some labs, patients that may not link correctly and temporal differences in availability of testing over the study period that may bias the results.

Thank you for this, we have extended the methods sections referring to this linkage and added a comment about testing linkage and biases to the limitations, and added some more context in the discussion.

Page 5 Line 22 – same as above. A top line reporting of quality of linkage/missing data would be helpful.

Thank you for this comment, we couldn't find published literature describing the quality of the linkage, but we have cited the technical reports from UKBiobank and added a comment.

Page 5, Line 44-45 – not sure that the approach of unidirectional state transitions gives a “more detailed” overview but this is nitpicky. Consider reframing to simplification of the model or emphasizing that the outcome of interest was just the first transition into any of these states.

Thank you for the suggestion. We have revised the sentence removing the reference to “detailed overview”, and indicating that the outcome of interest was just the first transition, as suggested.

Page 5, Line 52-53 – missing exposure data is presented in the study flow diagram. What about missing data from the outcomes (patients that did not link into other databases).

Because the dataset used is restricted to England we do not expect missing data from outcomes, as UKBiobank is fully linked with HES.

Page 6, Line 6-7 – covariates in the final models are listed but prefer if this paragraph would have some justification for inclusion of each variable. Could be done with references or expert opinion as the included covariates are the usual ones. However, some explanation of territorial measure of socioeconomic deprivation versus individual data should be detailed here. Its not clear why this measure was used instead of an individual measure (was it just not available in the existing data or linked databases?). Also, given the regional variability of surges in Covid-19, was there consideration of a model that includes a variable to capture regional prevalence (within England) as this may greatly influence all outcomes. Was there consideration of other variables such as comorbidities, primary language other than English, immigrant status or other variables known to influence healthcare utilization by ethnicity or race.

Thank you for this comment. We have added a reference to justify the covariates added to the model. We have integrated comorbidities into the model; the remaining suggested variables are not available in the database. We added a line to the discussion section to indicate that further research should consider additional covariates.

Page 6, Line 11-22 – This section should be expanded significantly with definitions of each outcome explicitly stated. Which Covid tests counted as positive/negative tests. Report LOINC. Please make it clear how hospitalization “with covid” was defined. Were only hospitalizations where Covid was the primary diagnosis used or were all hospitalizations used as the outcome? Same with 28 day mortality – please clarify whether or not Covid was required to be primary diagnosis or reason for hospitalization.

Under outcomes we indicate that “COVID-19 related death or hospitalisation with COVID those which occurred within 28 days of a positive SARS-CoV-2 test”. We have added a sentence to indicate that COVID-19 was not required to be the primary diagnosis or reason for hospitalisation. We have also added that “whether tests were positive or not was obtained directly from Public Health England as described elsewhere (6)”.

Page 6, Line 30-34 – would be helpful (maybe in supplemental) to provide the exact model specifications for each transition modeled as it seems like there will be both left and right censoring, maybe a supplemental table of all the models, censoring, and maybe some notes about fit testing. Its not needed in the main text but would be informative as supplemental. Could consider reporting bivariate associations here too such as median time to (transition state from prior) with Cis.

Thank you for this comment, we believe this could be useful, but it was not a standard output of the models and we don't have access to the raw data anymore, as the UKBiobank project finalised, so we weren't able to add this information.

Page 6, Lines 35-36 – this appears to be an analysis of an outcome, number of tests, that is not described above in the outcomes section. Please report how this was defined above.

We have added to the outcomes section in methods the description of the number of tests analysis.

Page 6, Statistical methods section – please report how model assumptions were tested and if any other model diagnostics were performed. Were any interactions assessed? I recommend doing this as a sensitivity analysis because in other health disparities, gender-race interactions or race-age interactions are common. There could be others. If no interactions were evaluated that's ok, the group sizes are very large but should be reported as a limitation.

We thank the reviewer for this comment. The underlying proportional hazards assumption of the Cox model employed here was tested through Kaplan-Meier estimator plots. Survival curves showed that hazard between groups is sufficiently proportional to justify the use of this model. Additionally, confidence intervals were the main indicator of statistical significance. It pointed to those transitions in which there are not enough observations of an occurrence to correctly assign risk to a group feature. Regarding covariate interactions, we didn't evaluate interactions so we have added a sentence in the limitations section.

Results:

Page 7, Line 15 – the outcome counts for the non-white groups are very low (single digits) so its difficult to draw any real conclusions from these groups and should be discussed as a limitation later in discussion.

Thank you for this comment. We have added a sentence in the limitations.

Page 8, Line 5-11 – 4.5% test positivity rate seems about right relative to general UK population at 18 months into the pandemic. However, death rate (~3% with 662/18,172) seems high suggesting overrepresentation in the numerator. Please explain in the discussion.

That is an important point, indeed, probably driven by the much older population in UKBiobank, and we have added a paragraph in the discussion.

Page 8, line 16 – in figure 3, the forest plot, please include an X axis. Its difficult to tell if its linear or log scale and this would inform the take away message from the size of the error bars.

We thank the reviewer for this comment. Unfortunately, the software we used to generate this table does not support showing the axes. We have added an indication that the x-axis is shared, linear, and has a range of [0,3] where the vertical line represents a value of 1.

Discussion:

Page 11, Line 37-41 – Discussion of weaknesses should include interpretation of how differences between Biobank and general UK population may affect generalizability of results. Also, do these differences (biobank vs general population) occur in ethnic/racial minorities? Some discussion of this would inform the reader’s interpretation of results as well. For example, are higher socioeconomic status Asian’s overrepresented relative to the general population of Asians in the UK?

The paper which interrogated differences between the UK Biobank and the UK general population (Fry et al, 2017, which we cite) did not examine the socioeconomic breakdown of each ethnic group included in the UK Biobank compared to the general UK population so it is difficult to reach any conclusion. It is very likely that all the population represented in UKBiobank is more well off, also inside ethnicity groups, and that this has an impact on the results.

Page 11, Line 41-44 – Please address how increasing availability of home tests during the study period influences the results. For example, do white participants have a lower risk of testing positive because they are home testing and not reporting the result?

We have added expanded the paragraph in the discussion about barriers and facilitators of healthcare among ethnic minorities and how this could have affected every step of the multistate model.

Page 12, Line 12-28 – The authors acknowledge that the reasons for racial and ethnic disparities in Covid outcomes are complex yet speculate about shift work as a specific reason for the observed disparities in this study. To me this seems out of place and a broader summary existing evidence about covid outcomes disparities would be more appropriate. The manuscript does not discuss or present data regarding shift work or key workers at all and only makes the connection in the discussion.

Many thanks for this suggestion. We have removed the reference to shift work and added the following more general statements about employment and wider determinants, to complement the points made on income and discrimination.

Page 12, Line 29-37 and 43-46j – This discussion of systemic biases in health care is pertinent and perhaps highlights potential next steps in this work? For example, it seems possible from the existing datasets used in this study to investigate availability of testing and vaccination by territorial region which is a testable and intervenable source of systemic bias. Would love to see proposed future analyses such as this in this section.

Many thanks for the suggestion. We have added a sentence indicating that “Further research using the datasets from this study, for example, could investigate availability of testing and vaccination by territorial region, which is a source of systemic bias.”

VERSION 2 – REVIEW

REVIEWER	Fung, Christopher University of Michigan
REVIEW RETURNED	31-May-2023

GENERAL COMMENTS	Thank you for the opportunity to review the revised manuscript entitled “The association between ethnic background and COVID-19 testing, status and health outcomes: a multi-state cohort analysis”. This is a large population-based cohort study leveraging the UK Biobank’s half a million participants. The study authors utilized this existing longitudinal study to follow participants and track their time of transition between various Covid health states in a multistate model of Covid-19 testing and infection. Participants were followed from April 2020 to September 2021 which is a period that roughly includes the first major surge of Covid-19 infections in the UK at the end of 2020 and the beginning of another major surge of infections at the end of 2021. The goal of this study was to determine the association between ethnic background and various states (receiving first test, testing positive, hospitalization, vaccination and death) of health related to Covid. The authors report a primary finding that Black and Asian participants were more likely to be tested for Covid, more likely to test positive and more likely to be hospitalized. White and Asian participants were more likely to be vaccinated. Previously, I had recommended acceptance with minor revision with some minor changes regarding framing and possibly linkage to individual data if available rather than using population data as a covariate in the study’s regression models. The authors report that the raw data is no longer available to perform changes to analyses which is reasonable and sufficiently explained in the limitations. As the authors have commented on in the discussion, a significant limitation of this work is the sampling bias of individuals in the Biobank cohort – which likely overrepresents White and elderly patients. This likely limits the generalizability of this study’s findings. Despite this limitation, the study is well written and adds to the growing body of work suggesting inequity in Covid related outcomes during the height of the pandemic. Given the revisions which include further discussion of next steps, context of the findings and a robust discussion of limitations, I recommend acceptance of this manuscript. Thank you again for the opportunity to review for BMJ Open.
---

VERSION 2 – AUTHOR RESPONSE

Please address the following comment from reviewer 1 in the manuscript:

Page 7, lines 12-14: Please indicate if the quantiles of the Index of Multiple Deprivation were defined by an identical (one-fifth) number of citizens in each quantile, otherwise the statement that “the upper quantiles of socio-economic status were over-represented” might be inaccurate. Not every national measurement system divides the quintiles or quartiles into an identical or almost identical numbers of individuals in each category.

We have reworded that sentence to make it more descriptive, without judging representation:

“Participants living in areas in the upper quintiles of socio-economic status were more frequent in the cohort overall (29.7% in Q1 vs 12.8% in Q5)”

UKBiobank comments

Could you please move the standard acknowledgement with the application number to the ‘acknowledgements’ section? Additionally, please can you add the two new acknowledgements required for use of linked health data.

I have added the acknowledgments.

Please can you also amend your tables to state <5 where the numbers of individuals cross-tabulated is less than 5.

I have masked the <5 in the descriptive tables.